# An Optimization-Based Approach to Evaluate the Operational and Environmental Impacts of Pick-Up Points on E-Commerce Urban Last-Mile Distribution: A Case Study in São Paulo, Brazil

Rhandal Masteguim * and Claudio B. Cunha 

Graduate Program in Logistic Systems Engineering, Escola Politécnica, Universidade de São Paulo, Sao Paulo 05508-220, Brazil; cbcunha@usp.br
* Correspondence: rhandal_fm@hotmail.com; Tel.: +55-(11)-99441-9741

**Abstract:** Online sales have steadily increased in recent years. Unlike the traditional retail shopping model, e-commerce must deliver custom orders to highly dispersed locations. Consequently, negative effects have been observed in large urban and densely populated areas, such as congestion and pollution. Pick-up points (PPs) are fast-growing solutions that provide parcel delivery and picking services at diverse locations throughout major city centers. This paper describes an optimization-based approach aimed to investigate the conditions in which a network of pick-up points can be more efficient than home deliveries from operational and environmental points of view in urban last-mile distribution. Differently from the related literature, in which analytical models were employed, we use optimization and algorithms to determine the economic and environmental benefits of packages destined for pick-up points instead of home deliveries. The framework was applied to the city of São Paulo, in Brazil. Several scenarios were evaluated, comprising different densities and percentages of deliveries destined for PPs. The results show that PPs can be a promising alternative for reducing the environmental externalities, as fleet and vehicle mileage can be reduced by more than 50%.

**Keywords:** last-mile distribution; e-commerce; pick-up points

## 1. Introduction

In the recent years, e-commerce has become one of the most important sales and distribution channels in the world, surpassing even brick-and-mortar retail purchases, as the variety of products that can be commercialized has greatly expanded, ranging from perishable foods to furniture. Its growth has been boosted by the recent global COVID-19 pandemic, which has imposed changes in consumer habits as circulation restrictions in various parts of the world have made online shopping either the only or the safest alternative.

Online sales have been steadily increasing, and around 95% of all purchases are expected to be via e-commerce by 2040 according to Nasdaq's research [1]. Better prices, greater product selection, convenience, and time savings are among the main reasons for such growth [2]. Projections from Euromonitor International show that about USD1.4 trillion corresponds to goods sold online, which is equivalent to 50% of the absolute value growth for the global retail sector between 2020 and 2025 [3]. Globally, China and the United States will account for 55% of that value growth, while Latin America registered a 60% growth in e-commerce sales in 2020.

Hence, last-mile distribution has become more fragmented. Contrarily to the traditional brick-and-mortar shopping model, in which customers do most of what is commonly referred to as "the labor-intensive work" (such as order picking and transporting the goods home), in e-fulfillment, logistics operators are facing the challenges of making deliveries increasingly fragmented and scattered within restricted delivery windows [4]. In other

words, the vast majority of the online orders are individual shipments of products to be delivered to consumers at their homes.

Consequently, last-mile delivery has become one of the key challenges for the expansion of e-commerce, due to several reasons, ranging from delayed delivery times to return of the parcels due to unavailability of the customer; they also constitute the most complex and costly operation in the supply chain [5], not to mention the externalities that arise, such as street and curbside congestion and pollution, especially in large, densely populated urban areas. The negative effects on the environment, particularly greenhouse gas (GHG) emissions, are intensified as e-commerce customers usually prioritize cost and speed over sustainability, especially in the last-mile distribution of purchased goods [6].

Although home delivery is preferred by online shoppers, new delivery alternatives have arisen that allow the packages to be picked up from convenient locations, such as pick-up points (PPs), which may include automated parcel stations (APS) equipped with lockers (also known as parcel lockers). Both options are playing a significant role from both commercial and logistics points of view. In the United States, Amazon has decided to invest in their own APS solution, while in France, more than 20% of the online shopping shipments are delivered through some pick-up point (PP) as 90% of the French population has access to a PP that is less than 10 min by car or on foot [7].

In this context, this paper describes an optimization-based approach to investigate under what conditions pick-up points (PPs), which can be either of a brick-and-mortar type or automated parcel stations (APS), may be more efficient than home delivery from operational and environmental perspectives in the context of large cities. An integrated framework was developed aimed to allow the evaluation of distinct delivery scenarios in terms of total deliveries and the percentage of deliveries destined for lockers. We also describe the results of its application in the context of the megacity of São Paulo, Brazil.

The remainder of this paper is organized as follows. The next section presents the literature review, followed by a description of our proposed approach. We then present the results. The concluding remarks are in the last section.

## 2. Literature Review

The literature on impacts of e-commerce last-mile delivery to pick-up points and other package retrieving alternatives for e-commerce home deliveries is still scarce, as the growth of online shopping is relatively recent. Most of the few studies found in the literature have focused on the factors that contribute to the location and usage of such facilities. In one of the pioneering works, Morganti et al. [5] analyzed the key drivers that led to the adoption of pick-up points (PPs) in stores and automated lockers (APS) by parcel delivery providers in France and Germany. A framework was proposed to identify the key aspects related to the design of a PP network, which was then validated by a survey with the parcel operators and storeowners. Through a case study in a large area adjacent to Paris (France), they found that the average distance to the nearest PP was 1.6 km and the travel time to the nearest PP was less than 10 min by car for 91% of the population; additionally, PPs tended to be located in main commercial streets. Iwan et al. [8] analyzed the usability and efficiency of parcel lockers in the last-mile distribution for InPost Company in Poland. They conducted a pilot survey with users of parcel lockers in Szczecin (Poland), whose results showed that the price of deliveries as well the location of such facilities in terms of convenience (i.e., close to home, on the way to work, or where it is easy to park for retrieving packages) were the most important reasons for their use. In a related work, Lemke et al. [9] analyzed the use of parcel locker services provided by InPost and other courier companies in Poland from a customer perspective. A survey with nearly 3000 respondents evidenced a high degree of satisfaction with such services, regarding them as better than couriers and even the national Polish Post. Price and the possibility of retrieving packages at any time were also seen as competitive advantage, as well as the availability of station locations in the vicinity of customer homes, shopping malls, and public transport stops, as they can be reached either on foot or while traveling to/from work.

While these studies related to PPs provided some relevant aspects related to their ideal location and distances from a customer's point of view, operational and environmental aspects and potential benefits of destining packages for stations were not measured or quantified. Recently, Saad and Bahadori [10] compared the sustainability performance between home service delivery and pick-up point service delivery using parcel lockers. Their results show that the pick-up point alternative outperformed the home service delivery for distance traveled (43% reduction) and $CO_2$ emission (39% reduction).

Ghajargar et al. [11] presented the results of a small survey to identify the habits, requirements, and perceptions of e-commerce home delivery services from the point of view of the users of such services. Key outcomes comprise the perception of such delivery services as low cost, as well as the benefits of automated pack stations in terms of increased level of service as customers may receive same-day delivery at the same price as that of the standard delivery as consolidation and fewer failed deliveries allow a more efficient delivery operation.

Based on interviews with logistics providers, Cardenas et al. [12] proposed an e-commerce delivery analytic model that allowed the comparison between home deliveries and the use of pick-up points. The results showed that pick-up points can yield to distance and cost reductions as a large percentage of the total number of parcels are consolidated using a relatively low number of pick-up points; in addition, such benefits are reduced in the case of larger networks of pick-up points.

Melkonyan et al. [13] developed a toolset for examining the sustainability potential of last-mile delivery and distinct distribution strategies (i.e., centralized versus decentralized distribution network with click and collect and home delivery alternatives, respectively, as well as a distributed network based on crowd sources) applied to a local food network. The toolset comprises a dynamic simulator that allows modeling future interactions among the system's relevant elements and a multi-criteria decision tool (to allow weighting such elements). The results evidence that distributed networks relying on crowd logistics perform better for local food networks from a sustainability point of view in comparison with the other two alternatives. However, their approach does not allow simulating different network configurations in terms of the number and locations of the facilities.

Rabe et al. [14] proposed a simulationoptimization approach with the aim to determine the best implementation plan for automated parcel lockers (APLs) in the context of last-mile distribution, comprising a system dynamics simulation model (SDSM) and a multi-period capacitated facility location problem (CFLP). The SDSM is used to investigate the behavior of the components of APL systems with respect to the specific customers and characteristics of city selected for the case study (Dortmund, Germany), while the multi-period CFLP determines the optimal number and location of APLs to be installed monthly, for a 3-year time horizon. In order to address demand uncertainty, different scenarios (e.g., e-shopper rates and demand configurations) were considered and solved and the respective solutions were then submitted to Monte Carlo simulations to allow estimating both their costs and reliability levels. The results evidence that the optimal number of APLs shows a linear behavior with respect to the potential users of APLs, with no obvious scale effects. Differently from the present work, demand was considered at an aggregated, district level; also, transport costs were considered in a simplified manner (i.e., unit costs between districts where an APL and a customer are located), not taking into account the distances and costs of delivery routes. Other works are related to the impacts of e-commerce versus traditional shopping. Edwards et al. [4] compared the $CO_2$ emissions between online and conventional shopping (i.e., that requires travel to establishments) from a last-mile perspective for small non-food items (e.g., books, CDs, electronics, and apparel). Their results show that it would be necessary to buy at least 24 or 7 items to offset the $CO_2$ emission of a home delivery if the shopping trip is made by car or using bus public transport, respectively. Effects of different rates of failed home deliveries and unsuccessful shopping trips (for instance, in case the customer end up not buying anything) were also analyzed. They conclude that, in general, home deliveries generate less pollution than conventional shopping; however,

they also highlight that they did not consider that such trips can have social or recreational motivation nor that they can be combined with other purposes on a single trip.

Durand and Gonzalez-Feliu [15] analyzed the impacts of e-grocery on household shopping trip behavior. Store, warehouse, and depot picking, as well as home deliveries, were compared and the results evidenced that consolidation of home deliveries and proximity reception points (where most trips are made on foot) can lead to a significant reduction in motorized distances traveled. Gonzalez-Feliu et al. [16] proposed a modelling framework to estimate the number of trips and traveled distances for shopping drive, home delivery, and reception points strategies. Results from different scenarios from the city of Lyon (France) showed that a combination of shopping trips (50%), home deliveries (15%), and proximity depot picking (35%) yields the maximum reduction (13%) in road occupancy. Other aspects related to how city logistics can help address the impacts of e-commerce deliveries can be found in Nemoto [17] and Hayashi et al. [18].

Gatta et al. [19] evaluated the economic and environmental impacts of an environment-friendly crowdshipping service based on public transport for the city of Rome, where customers and crowdshippers (i.e., passengers that use the transit system) can retrieve and/or deliver parcels in automated parcel lockers located either inside or in the vicinity of transit stations. Based on a stated preference survey to determine the potential demand (i.e., willing to adhere to a crowdshipping service), many different scenario analyses were performed using discrete choice modeling to calculate revenues for the operator, required investments, and operational costs for the crowdshipping platform and externalities as well. The results show that implementing such a service in Rome may yield total yearly savings of pollutants (1089 tons of $CO_2$, 3.76 tons of NO, 2.24 tons of CO, and 239 kg of particulates).

Finally, Kiba-Janiak, Marcinkowski, Jagoda, and Skowrońska [20] provided a systematic literature review of sustainable urban last-mile delivery for the e-commerce segment considering the perspectives of different stakeholders. Using both traditional method and machine learning, the authors found out that the largest group of papers relate to sustainable last-mile delivery and address the receivers' perspective (i.e., consumer habits, shorter delivery times, and pick-up point planning). The authors also found that more recent articles address technological and organization aspects, which also include parcel lockers.

There is a large and growing body of literature related to the different types and variants of the vehicle routing problem (VRP); we refer the reader to Braekers et al. [21] for a recent taxonomic review and classification of the academic literature on the VRP. Cattaruzza et al. [22] surveyed the vehicle routing problems found in cities for good distribution and identified the important challenges related to routing in urban goods distribution as well. With respect specifically to vehicle routing related to e-commerce delivery, the work from Archetti and Bertazzi [23] focuses on routing and inventory routing variants that arise as online orders require fast delivery, leading to characteristics that may differ from classical routing problems. Among additional relevant related works, Moons et al. [24] studied the joint problem of order picking and routing vehicles with time windows and release dates. Computational experiment with the proposed MIP formulation showed that such integration can lead to cost savings of 14% on average with respect to an uncoordinated approach; however, it comprised only very small instances (10, 15, and 20 orders).

Özarik et al. [25] studied a variant of the VRP in which the aim is to minimize the sum of the routing cost and the expected time-dependent penalty cost resulting from missed deliveries that derives from customer availability profiles. The problem is modeled and solved using an ALNS search metaheuristic that iterates between the routing and scheduling components. Computational experiments using data from benchmark instances show that savings up to 40% can be achieved compared with VRP traditional solutions.

Vincent et al. [26] addressed a new variant of the vehicle routing problem with time windows (VRPTW), named the vehicle routing problem with parcel lockers (VRPPL), by adding locker delivery as one of the delivery options. More specifically, it involves three types of customers to be served: those whose packages must be delivered to their respective

domiciles within predetermined time windows; those whose packages need to be delivered to given parcel locker stations of their preference, and those who can receive packages in both ways. The goal is to minimize the total distance traveled by all vehicles subject to time window constraints (for home deliveries) and vehicle and parcel locker capacities. A new mathematical programming model and a simulated annealing (SA) algorithm are proposed to solve it. Randomly generated datasets based on benchmark VRPTW instances with up to 100 customer nodes were derived to evaluate the performance of the SA algorithm. The results showed that the heuristic approach outperforms the exact method for the largest instances. However, it should be noted that instance sizes are still modest when compared to those found in practice. Additionally, the exact modeling could be enhanced by using a more efficient approach, such as column generation.

To conclude, it is interesting to observe that the extant literature related to the comparison of e-commerce home delivery with PPs but also with traditional shopping and other alternatives, such as crowdshipping, relies mostly upon analytical models. To the best of our knowledge, this is a novel optimization-based approach, comprising a routing algorithm to analyze and compare PPs (either of a brick-and-mortar type or APS) with home deliveries from operational and environmental perspectives, in the context of large cities, using realistic data, comprising different densities of deliveries, locations of the origin fulfillment center, and percentage of deliveries destined for lockers. In addition, the size of problems we address, measured in terms of customer orders, is significantly larger than those usually found in the literature.

## 3. Problem Description

Our aim is to investigate under which conditions pick-up points (PP) can be more efficient than home deliveries from operational and environmental perspectives. For this, we propose a framework that mimics a real operation of distribution of e-commerce products focusing on the context of large cities in developing countries. Two distribution alternatives can be simulated: parcels being delivered at the residences of the buyers or at pick-up points. Our optimization-based framework is designed to determine optimized solutions for both distribution alternatives as well as the comparison of performance and efficiency between the operations, in terms of the number of vehicles used, mileage covered, fuel consumed, and GHG emitted.

The problem comprises a B2C (business to consumer) e-commerce last-mile delivery system in urban centers (Figure 1). We are given a number of home deliveries to be performed in a predefined region called the "delivery region" on a single day. The geographic location of each delivery is known, as well as its drop size in terms of both weight and volume.

A heterogeneous fleet of urban small delivery vehicles of different sizes and types is available to perform the deliveries. Routes are subject to vehicle capacity (weight and volume) and time duration constraints. Vehicles are dispatched once a day and make only a single trip each, regardless of their duration. No backlogs are allowed; in other words, all scheduled deliveries are performed on each day using the best mix without any constraints on the number of vehicles of each type that may be required. Delivery time windows are not considered; similarly, urgent deliveries that impose a maximum time to deliver (such as, for instance, a two-hour delivery that is offered by e-commerce providers such as Amazon in some countries and locations) are not considered either, as it is unlikely that such deliveries would be destined for parcel lockers to be collected by the customer later.

All vehicles depart from and return to the origin fulfillment center (also referred to as depot), whose location, simply given by the distance to the delivery region, is known in advance. This allows evaluating the impact of different locations, farther from or closer to the delivery region. Typically, the greater the distance between the depot and the area to be serviced, the less time is available to perform the deliveries.

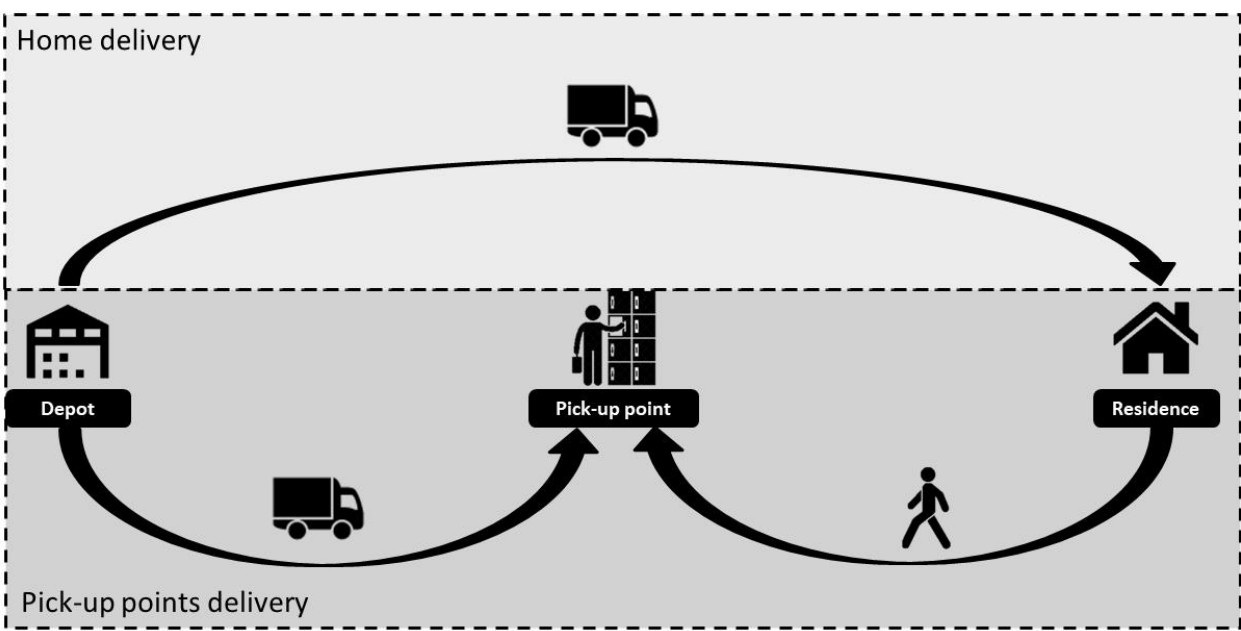

**Figure 1.** Problem representation.

A given proportion of home deliveries may be directed to potential pick-up points (of either the brick-and-mortar type or APS). The number and locations of such pick-up stations is determined within our framework among a pre-selected set of candidate locations that may comprise gas stations, supermarket stores, malls, transit terminals, and other public places, following the results by Lemke et al. [9]. All are capable of operating as a brick-and-mortar PP or an APS. In our framework, we also defined a maximum walking distance to a PP that is used when determining the optimal number and their locations.

### 3.1. Proposed Approach

In order to accomplish the objective of this study, we propose a comprehensive framework that comprises generating demand for standard home delivery operation as well as an operation using PP infrastructure in different demand patterns and optimization of the PP locations and vehicle routes as well. It is crucial to have comparable scenarios where the same demand is met, but in one scenario only uses home delivery operation and in the other scenario uses PP infrastructure. Therefore, it is possible to compare the total distance traveled and pollutants emitted and, thus, their efficiencies.

To summarize, the framework is aimed to allow the evaluation of distinct scenarios that may vary in terms of total deliveries and percentage of deliveries destined for lockers, yet allows maintaining operational definitions, such as customer maximum walking distances to collect a parcel (which defines the number of PPs), fulfillment center location from where the vehicles depart, and fleet operational parameters. It comprises five stages, as depicted in Figure 2.

In Stage 1, different demand alternatives are set: the number of packages to be delivered and for each package, its weight and volume and geocoded destination location.

In the second stage, our framework determines the optimal number and location of PP candidates needed to meet the demand destined for them. For this, a linear programming model for locating installations is used, whose objective function is to minimize the number of PPs needed. This model considers a pre-defined set of PP candidates as decision variables in order to meet the demand, also considering the restriction of the maximum one-way walking distance from their original addresses and the minimum percentage of demand that must have at least one locker option available for delivery within the maximum walking distance.

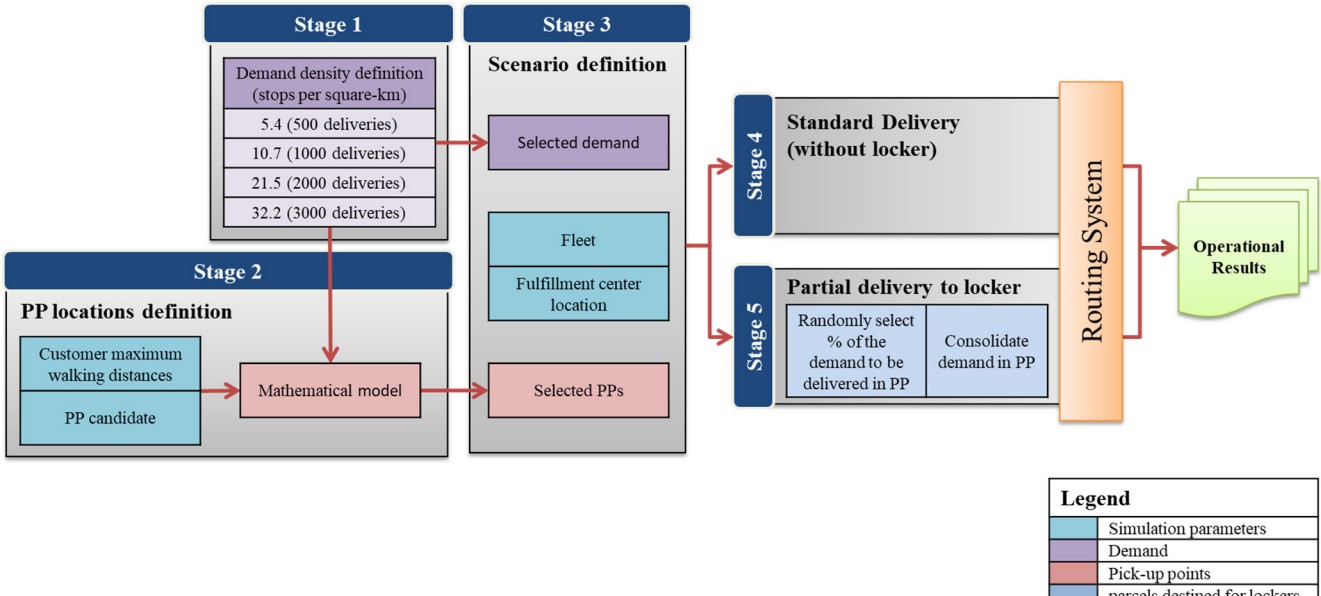

**Figure 2.** Proposed procedures scheme of the framework.

In Stage 3, we configure each scenario: a given demand pattern, the available fleet in terms of vehicle types and their operational parameters, and the locations of the PPs. Each change in one or more variables in this step corresponds to a new scenario.

For each set of defined parameters (scenario) in Steps 4 and 5, different scenarios are evaluated that differ only in the amount of demand delivered by the PPs. The first simulation occurs in the fourth step, which represents the conventional home delivery operation without using PPs, and the operational results from this simulation will be the basis of comparison for the following simulations.

In the fifth stage, scenarios considering PPs take place. Different percentages of the demand will be delivered to a PP location instead of to their original addresses. In addition, at this stage, the packages that result in the desired percentage of deliveries destined for lockers are randomly selected and each of these is allocated to its respective nearest PP.

Once the fifth stage is completed, the original addresses are no longer part of the distribution routes and the selected PPs become the new delivery locations, consolidating deliveries into a smaller number of stopping points.

### 3.1.1. Mathematical Model

A mathematical model is proposed to determine the optimal number of PPs/APS and their locations. A set-covering integer programming facility location optimization model was initially proposed to determine the minimum number of PPs to be selected, as well as their locations, to ensure that all delivery points are within the maximum walking distance [27]. However, after applying the model and analyzing the preliminary results, distortions were identified that would not represent the real operation of a logistics operator, such as the opening of PPs in regions with low demand for a single package. This distortion makes the set-covering model unsuitable to represent a real operation, because this model is restricted to meeting the total demand, that is, all packages must have at least one PP within the maximum one-way walking distance.

In order to solve the distortions described above and represent the operation in a more realistic way, the following changes were proposed in the set-covering model:

- Inclusion of the restriction of the minimum number of packages to open a PP;
- Flexibility in the restriction of demand fulfillment with the inclusion of a parameter that represents the minimum percentage of the total demand that must have at least one locker option available for delivery within the maximum walking distance.

The model with the changes can be formulated as follows:

*Parameters:*

$m$ = Number of home delivery points

$n$ = Number of PP candidates

$d$ = Maximum walking distance between home delivery points and the PPs

$d_{ij}$ = The distances between home delivery point $i$ and the candidate PP locations $j$

*MinQty* = Minimum number of home delivery points destined for a single PP

*MinPerc* = Minimum percentage of total home delivery points that can be served by PPs

The distances between all home delivery points $i$ and the candidate PP locations $j$ were determined using Google Maps API.

*Variables:*

Let $x_j$ be the binary decision variables that take the value 1 if a PP location $j = 1, 2, \ldots,$ $n$ is selected and zero otherwise. Let $y_{ij}$ be the binary decision variables that take the value 1 if the home delivery point $i$ is destined for the candidate $j$ and zero otherwise. Let $z_i$ be the binary decision variables that take the value 1 if the home delivery point $i$ is destined for any candidate and zero otherwise.

*Model formulation:*

Minimize

$$\sum_{j=1}^{n} x_j \tag{1}$$

$$\text{Subject to}: \sum_{i=1}^{n} z_i \geq MinPerc \cdot m \tag{2}$$

$$\sum_{j=1}^{n} y_{ij} \geq z_i, \ \forall i = 1, \ldots m \tag{3}$$

$$x_j \geq y_{ij}, \ \forall i = 1, \ldots m, \ \forall j = 1, \ldots, n \tag{4}$$

$$\sum_{i=1}^{m} y_{ij} \geq MinQty \cdot x_j, \ \forall j = 1, .., n \tag{5}$$

Objective Function (1) aims to minimize the number of PPs to be opened, while Constraint (2) ensures that the number of home delivery points destined for PPs is greater than or equal to a given percentage of the total number of delivery points. Constraint (3) ensures that if there is a service flow between a PP and a home delivery point, that point will be counted as serviced. Restriction (4) guarantees that there will only be a service flow between a PP and a home delivery point for PPs that are opened, and Restriction (5) guarantees that a PP can only be opened if it has a minimum number of home delivery points being served by it.

### 3.1.2. Determining the Optimal Vehicle Types and Routes

A vehicle routing algorithm was needed to determine the types of vehicles to be used and the routes that minimize the total delivery costs. The problem to be solved corresponds to a heterogeneous fleet sizing and routing problem with capacity and route duration constraints; no time window constraints are considered, as they are not common in e-commerce deliveries in Brazil or in other developing countries. It is a complex problem that cannot be solved to optimality by means of an exact formulation, thus requiring the use of heuristics to real-sized problems [28], as is our case. We initially considered using an available spreadsheet solver for vehicle routing problems (VRPs) [29], given the considerable time and effort required for developing an efficient routing algorithm producing high-quality results from scratch. VRP Spreadsheet Solver is an open-source Excel-based tool for solving many variants of the vehicle routing problem (VRP), including Capacitated VRP, with time duration constraints and heterogeneous fleet. However, the solver can solve VRP instances with up to only 200 customers, which is insufficient given the number of daily deliveries that have to be scheduled.

We thus developed a tailor-made heuristic to solve this problem. It comprises some different construction heuristics and some local search improvement heuristics, whose details are beyond the scope of this paper. The proposed heuristic was applied to 143 VRP instances from the benchmark set, aiming to evaluate the quality of the solutions it can obtain [30]. The results show an average deviation of 5.3% with respect to the total distance traveled.

Finally, it is important to highlight that the optimal number and location of the PPs is typically a tactical decision (i.e., can be reviewed periodically) that depends solely on ensuring the minimum coverage in terms of the maximum walking distances to PPs in order to retrieve packages for a given percentage of the domiciles in the selected area. In other words, such optimal configuration is not influenced by routing decisions, which are essentially operational in nature (as made on a daily basis). Thus, since there is no cost trade-off between location and routing, once optimizing the number and location of the PPs does not depend on routing aspects, these two types of decisions do not need to be integrated and can be made independently.

### 3.1.3. Routing Heuristics

Akin to other approaches found in the literature, the routing algorithm we have developed seeks the best combination of routes and vehicles, with the objective of minimizing logistical operations costs. It relies on classical methods, and algorithms described and extensively used in the literature generally produce good-quality solutions within modest computing times [28].

The following input parameters are required:

- Customers that need to be serviced (including their geocoded locations) and the amount to be delivered in terms of volume and weight.
- The location of the depot (DC) from where delivery routes start and end.
- Vehicle types that can used, which differ in capacities, costs, travel speeds, stop times (which may differ based on the difficulty to park), and maximum journey lengths.

The overall description of the heuristic is as follows.

As with Cinar et al. [31], we employ the K-means clustering heuristic to cluster the delivery points; however, differently from their work, we impose that the total amount (volume and weight) to be delivered in each cluster does not exceed the capacity of the smallest vehicle available in the fleet. Thus, the number of clusters is obtained as a result of our modified K-means heuristic instead of an input, as originally proposed in the K-means clustering heuristic (see, for instance, Cao and Glover, [32]). Then, for each cluster, an initial route is created using the nearest neighbor heuristic starting from the DC (beginning of the route) and going through all the demand points that comprise each cluster. These initial routes are improved by a simple improvement heuristic that comprises searching and removing intersections, as shown in Figure 3.

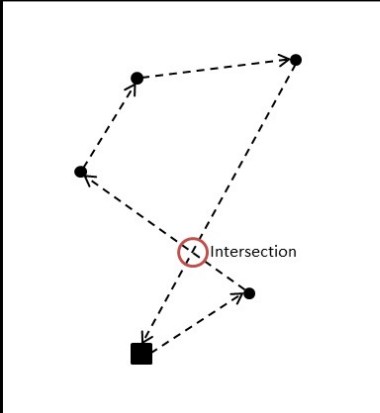
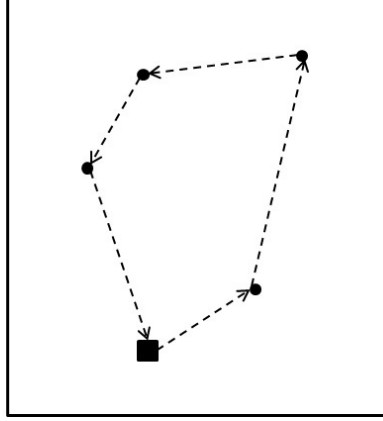

**Figure 3.** Intersection repair heuristic.

Once a feasible solution has been found, a local search heuristic is applied to improve the current solution. It comprises several local search methods, which include (i) a route combination heuristic that is used to delete routes and assign customers to the closest routes using a larger vehicle and (ii) swapping or relocating customers to different vehicles. For all changes made and new solutions found, a feasibility validation is performed, testing whether the vehicle's occupancy is less than its capacity and the total duration of the route is less than the working time limit for each route.

Finally, the incumbent solution is partially destroyed and then repaired again to overcome a local optimal solution in the following way: one route is randomly selected, and a given number of adjacent (neighboring) routes is also selected to be destroyed and then rebuilt by employing the modified K-means clustering, followed by the construction and improvement heuristics. This procedure is performed for a given number of iterations.

## 4. Data and Parameters Definition for the Case Study in São Paulo, Brazil

The data used for this case study come from the city of São Paulo and can be segmented into resources and environment parameters. Resources are parameters that affect the operational results and can be changed, that is, they are the parameters that differentiate each solution. For this analysis, we considered the demand density and percentage of deliveries destined for lockers as the resource for the framework; environment (or restrictions) is a parameter that affects operational results and cannot be changed. For this study, we considered fulfillment center location, fleet operational parameters, and PP candidates as environment parameters for the framework.

### 4.1. Demand Density and Percentage of Deliveries Destined for Lockers

We selected an affluent central and congested area of approximately 93 km$^2$ in the city of São Paulo, Brazil, as depicted in Figure 4. It hosts a population of 1.2 million people and was selected based on the high number of deliveries as well as its elevated population density (12.5 K people per square kilometer).

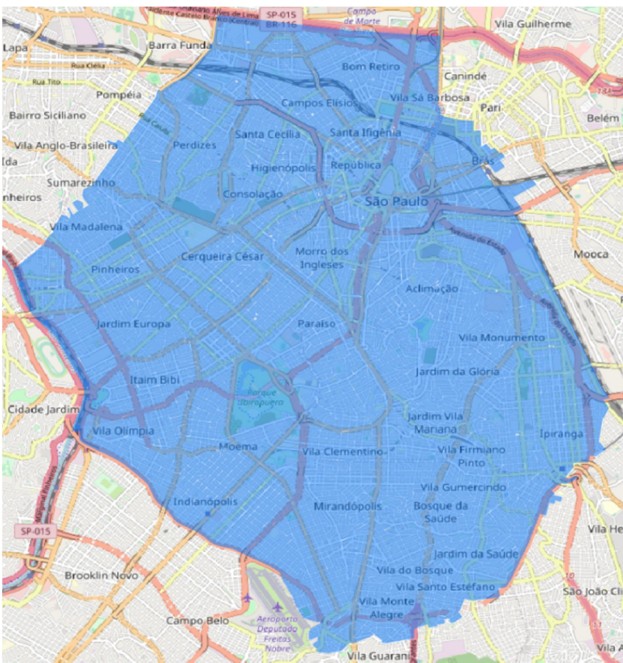

**Figure 4.** Area of study in São Paulo, Brazil.

In interviews with large e-commerce retailers in Brazil, an average of 1000–2000 deliveries per day was found as a typical range to the delivery area we have selected. One of the companies provided us with records of daily deliveries destined for the city of Sao Paulo for a period of time comprising 12 months. This dataset was subjected to data cleaning,

filtering, and processing, with the aim to obtain data for the selected region. Thus, in order to represent the reality of distinct e-commerce players as well as different demand patterns, we analyzed four distinct values for densities of daily delivery:

(a)    Five hundred deliveries, equivalent to 5.4 stops per square kilometer;
(b)    One thousand deliveries, equivalent to 10.7 stops per square kilometer;
(c)    Two thousand deliveries, equivalent to 21.5 stops per square kilometer;
(d)    Three thousand deliveries, equivalent to 32.2 stops per square kilometer.

Once the density of each district was established, we randomly generated delivery locations for several typical delivery days; they differ in terms of delivery locations (geographic coordinates) and drop sizes (weight and volume). All deliveries comprised small parcels.

A delivery location was checked in terms of geographical attributes, and those whose coordinates lay in parks, public squares, or not at roadside (i.e., in the inner part of a block) were eliminated and a new location was randomly generated

To generate the drop size of each delivery, we considered the weight and volume distribution of a major e-commerce retailer that provided a historical information data base, from which only the data were extracted to obtain the weight and volume distribution curves of possible dimension values. For our study, only deliveries weighing less than 9 kg and with volumes less than 0.05 cubic meters (50 L) were considered, as larger packages may not be easily accommodated in parcel lockers, not to mention the unwillingness of the customers to retrieve them instead of receiving them at their homes. In addition, local experience has shown that oftentimes, larger items are not handled and transported together with the typical packages that are covered by our study; not only they cannot be accommodated in the light delivery vehicles that we have considered but also they may require an assistant to the driver to perform the delivery. Figures 5 and 6 show the weight and volume distributions that we considered.

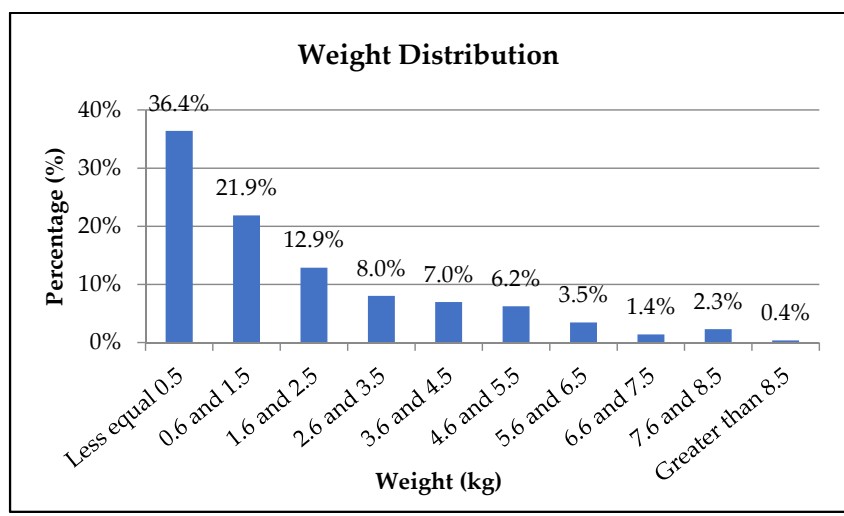

**Figure 5.** Weight distribution retrieved from data of a major e-commerce provider.

The percentage of deliveries destined for lockers determines the percentage of demand that will no longer be delivered to their original addresses but will be destined for lockers operation instead. This percentage is the other parameter that will vary to compose the different scenarios.

For each demand density, five scenarios were simulated associated with the percentage of parcels destined for lockers. The first scenario represents the conventional home delivery operation without using PPs, and the results from this simulation will be the efficiency comparison basis. For the following scenarios, four fixed parameters, corresponding to 20%, 40%, 60%, and 80% of demand, were defined to be destined for locker delivery, and

the parcels that resulted in the desired percentage were randomly selected and allocated to their respective nearest locker available.

## Volume Distribution

**Figure 6.** Volume distribution retrieved from data of a major e-commerce provider.

### 4.2. Fulfillment Center Location, Fleet Operational Parameters, and PP Candidates

Commonly, e-commerce deposits are located far from urban centers, given the characteristics of cost and accessibility. For this case study, the city of Cajamar, in the outskirts of Sao Paulo Metropolitan Region, was chosen as the fulfillment center location, for being widely known for housing fulfillment centers, commercial warehouses, distribution centers, and some manufacturing factories. It is located 40 km from the delivery region, of which 28 km consists of highways and the remainder is already part of urban traffic.

The heterogeneous fleet considered three different vehicles for the deliveries: a motorcycle courier, a light van, and a full-size delivery van. Note the inclusion of motorcycles in the fleet considered, as they are typical urban vehicles found in developing countries. As pointed out by Longhurst and Brebbia [33], motorcycles are being increasingly used as a means of transport, mainly in urban centers of large cities around the world, especially in developing countries, such as Brazil, India, and China. Such growth can be explained by their agility for the urban transport of small urgent items (such as meals), as well as lower fuel consumption, easy parking, and low acquisition and maintenance cost in comparison with other vehicle types (cars and vans). However, they are highly pollutant as they account for a significant part of CO emissions. Particularly in developing countries, urgent deliveries also rely mostly on motorbikes.

Table 1 presents the costs and operational parameters for the selected fleet considered. All costs are in Brazilian Reais (BRL).

**Table 1.** Fleet costs and operational parameters.

| Operational Parameters | Motorcycle | Light Van | Van |
|---|---|---|---|
| Fixed cost (BRL/day) | 110.21 | 174.79 | 223.36 |
| Variable cost (BRL/km) | 0.16 | 0.45 | 0.43 |
| Weight capacity (kg) | 30 | 650 | 1.540 |
| Volume capacity (m$^3$) | 0.135 | 3.2 | 10 |
| Workload (h) | 10 | 10 | 10 |
| Fixed time per stop (min) | 01:30 | 04:30 | 06:00 |
| Variable time per package (s) | 00:36 | 00:36 | 00:36 |
| Delivery travel speed (km/h) | 25 | 15 | 12 |
| Line haul speed (km/h) | 50 | 50 | 50 |

Fixed vehicle costs were calculated based on vehicle acquisition costs (BRL 11,085, 55,985, and 108,234 for motorcycles, light vans, and vans, respectively); they comprise the following components: depreciation (10% of the acquisition costs, yearly); capital costs (10% of the acquisition costs, yearly); insurance (20% of the acquisition cost for motorcycles and 7% for light vans and vans, yearly); and taxes (2%, yearly). Driver monthly costs are BRL 2477 for motorcycles and BRL 3192 for light vans and vans. The resulting monthly values were divided by 26 working days to find the daily equivalent cost.

The variable costs include fuel consumption, maintenance, and tire wear; they were obtained from the official specifications of the manufacturer of each vehicle. Considering the consumption of each type of vehicle (35 km/L for motorcycles, 10.7 km/L for light vans, and 10 km/L for vans) and the fuel cost (BRL 3.90 and BRL 3.20 per liter for gasoline and diesel, respectively), the resulting fuel cost per km was BRL 0.11 for motorcycles, BRL 0.36 for light vans, and BRL 0.32 for vans. The maintenance cost is 0.03/km for motorcycles, 0.06/km for light vans, and 0.08/km for vans. A motorcycle's tires have an average durability of 15,000 km, at the cost of BRL 350 for the pair, so the resulting cost per kilometer is 0.02. The tires of a light van have an average durability of 40,000 km and cost BRL 1000 for a set of four; so the resulting cost per kilometer is BRL 0.03. The tires of a van have an average durability of 60,000 km, with a cost of BRL 1800 for the entire set of tires, leading to a unity cost per kilometer equal to 0.03. The resulting fixed and variable costs are summarized in Table 1.

The capacities (weight and volume) were obtained from the official specifications of the manufacturer of each vehicle. Finally, the times (fixed per stop and variable per package) and speeds (delivery and linehaul) were obtained from historical real data from a major e-commerce retailer in Brazil.

These three vehicle types were selected as they are the most common ones used in urban parcel deliveries in Brazil and other developing countries. The operational and cost parameters were obtained from local parcel delivery companies. Note that the time to deliver comprises two components: a fixed time per stop and a variable time per package. The fixed time depends on the type and size of the vehicle; the time increases as a larger vehicle is used, thus reflecting the difficulty in finding a parking spot. In turn, the variable time is the same for all vehicle types. Similarly, travel speeds may change depending on the size of the vehicle. Motorbikes are faster as they are more agile and less subject to traffic congestion as they travel between traffic lanes when the traffic is snarling. For each vehicle type, there are two travel speeds: one when the vehicle in the delivery area is performing deliveries and the other for the segment between the fulfillment facility and the delivery area. As most of these facilities are located in the outskirts of large cities due to the logistics sprawl, highways or express roads with few traffic lights are majorly used to reach the more central areas. These speeds were estimated based on the feedback of the local parcel delivery companies and also on the results by Laranjeiro et al. [34]. Again, motorbikes can travel faster while the two other two-wheel vans have different and much slower speeds, which reflects their size.

With the aid of Google Maps, 606 real establishments were manually selected to be used as potential candidate locations for pick-up points in the selected area (Figure 7). As mentioned above, these locations comprise gas stations, supermarket and convenience stores, shopping malls, metro and transit terminals, among others. For the sake of simplicity and given the difficulty to survey actual data as well, we assume all PPs as equal, and their costs do not differ among locations within the region. Based on related literature, in particular the results from Lemke et al. [9], we adopted 800 m as the maximum one-way walking distance to retrieve packages.

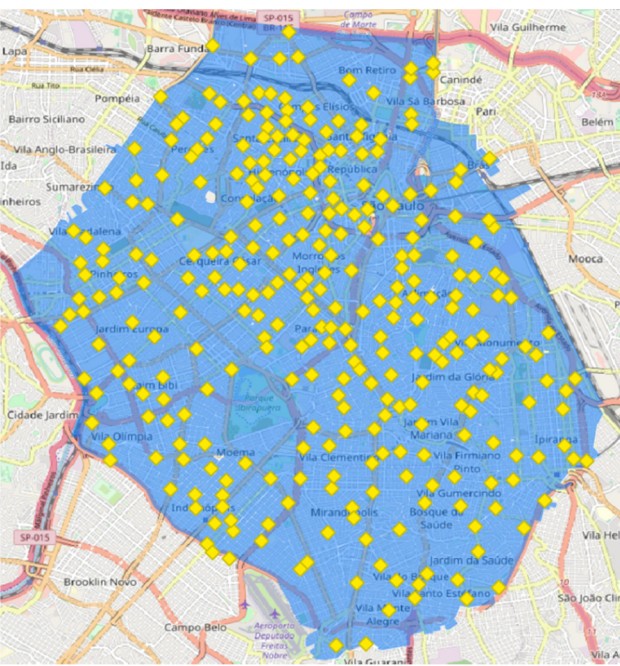

**Figure 7.** PP candidates.

## 5. Results and Discussion

Different alternatives have been evaluated comprising the analysis of several typical one-weekday home deliveries to be performed, each of them randomly generated as described in the previous section. The number of daily delivery parcels varies between 500 and 3000 different physical addresses. We also vary the percentage of deliveries destined for pick-up points. Four ranges have been considered: 20%, 40%, 60%, and 80%.

The scenarios were named considering the following nomenclature system: Q corresponds to the density of packages (demand) and A corresponds to the percentage demand delivered through the PPs, thus creating the following naming pattern: "Qxx.x_Ax.x".

The results of the scenarios described in the previous chapter are reported in this chapter and in Table A1 in the Appendix A. Each scenario corresponds to different conditions of use of PPs, comparatively showing the values obtained for vehicle–km and fleet size. Aspects of environmental impact are also addressed, such as the reduction in fuel consumption and the consequent emission of polluting gases.

### 5.1. Mileage (Vehicle–Km)

The reduction in mileage traveled is a direct result of the distances traveled in the delivery routes. Thus, the consolidation of deliveries in PPs (i.e., fewer stops) makes it possible to reduce the number of delivery routes as well as the distance traveled, measured in vehicle–km. Figure 8 shows the average percentage reduction in kilometers traveled for scenarios using PPs when compared to the home delivery scenarios. It should be noted that the percentage reduction curve of mileage seems to exhibit a linear increasing behavior.

It is possible to observe an average percentage of reduction in distance covered for each percentage (A0.2, A0.4, A0.6 and A0.8), whereby the average reduction in the distance covered with 20% demand delivered through the PPs is 11.1%; with 40% of the demand delivered through the PPs, it is 26.3%; with 60% of the demand delivered through the PPs, it is 41.1%, and with 80% of the demand delivered through the PPs, it is 56.8%. However, for each percentage of the demand delivered through the PPs, the mileage reductions associated with demand densities of 5.4 packages/km$^2$ are always significantly inferior when compared with the order demand densities. This behavior happens because, in these scenarios, the occurrence of PPs receiving only one delivery is more frequent, causing no load consolidation and, consequently, not significantly impacting the mileage traveled.

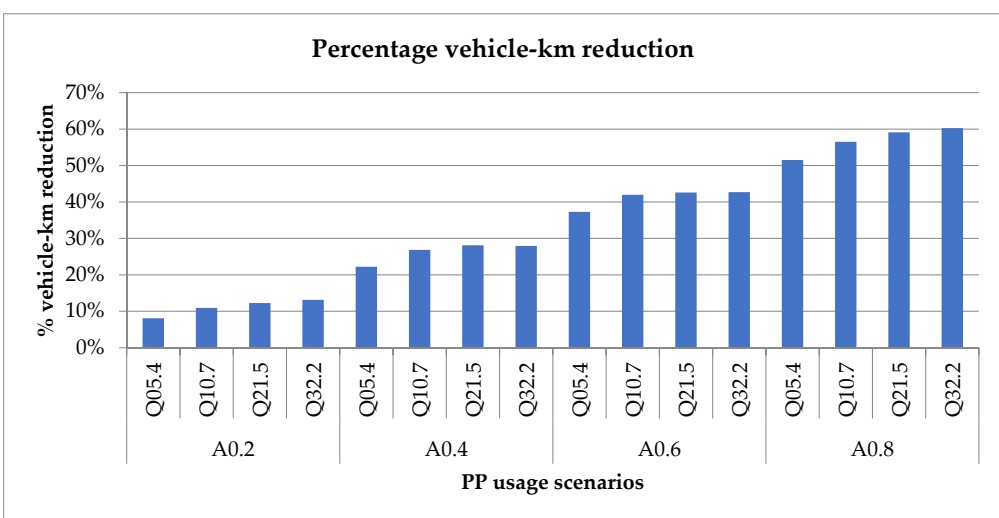

**Figure 8.** Percentage of vehicle–km reduction.

To illustrate how the consolidation of deliveries that are directed to PPs can yield a reduction in the distance traveled by the delivery vehicles, let us consider two distinct scenarios, Q05.4_A0.2 and Q05.4_A0.4, which differ only in the percentage of the 500 deliveries that are destined for 18 PPs: 20% or 40%. While in the former, there is a reduction of only 6% in the number of stops, which allows an 8% reduction in the vehicle–km, in the latter, the 318 stops (300 home deliveries and 18 at PPs) yield an average mileage reduction of 21% (i.e., approximately three times greater).

Table 2 summarizes the results we have obtained in terms of the total distance traveled (measured in vehicle–km) by all routes required to perform all deliveries in each of the scenarios. The operation in which PPs are used has made it possible to reduce to 64 km (home delivery compared Q05.4_A0.2) by an impressive amount of 2916 km (home delivery compared Q32.2_A0.8) in just one delivery day. Calculating this reduction for one year of operation (considering 26 days of delivery and 12 months in the year), the use of such retrieval points allows for a reduction of up to 909,000 km per year.

**Table 2.** Average total vehicle–km per day in each scenario (km).

| Density | Home Delivery | A0.2 | A0.4 | A0.6 | A0.8 |
|---------|---------------|------|------|------|------|
| Q05.4 | 999 | 935 | 793 | 609 | 453 |
| Q10.7 | 1800 | 1604 | 1318 | 1017 | 755 |
| Q21.5 | 3323 | 2910 | 2355 | 1824 | 1330 |
| Q32.2 | 4767 | 4087 | 3313 | 2572 | 1851 |

*5.2. Fleet Size*

The fleet size (the number of vehicles of each type required) is a result of the routing algorithm for each scenario (which aims to compose the lowest cost routes). As the number of orders/deliveries increases (Q), so does the number of vehicles required for the operation, as can be seen from the reference scenarios highlighted in orange in Figure 9, which correspond to the situation without PPs. However, note using an operation with pick-up points always allows cargo consolidation and reduces the number of vehicles needed. Figure 9 shows the reduction in the absolute number of vehicles used to perform the operation in each proposed demand (Q), with different percentages of the demand delivered by the PPs (A) to the pick-up point model. There is a pattern of reduction in the number of vehicles needed as demand delivered through the PPs increases in each of the scenarios.

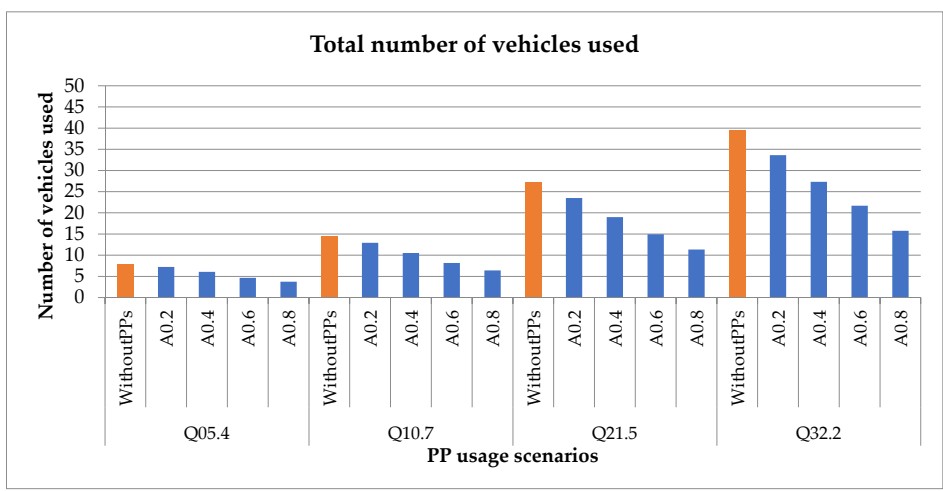

**Figure 9.** Total number of vehicles used.

The average reduction in the number of vehicles used as the demand delivered through the PPs varies is shown in Table 3.

**Table 3.** Average percentage of reduction in the number of vehicles.

| % Demand Delivered through the PPs | Average % Reduction in Vehicle Number |
|:---:|:---:|
| A0.2 | 12.4% |
| A0.4 | 28.3% |
| A0.6 | 43.8% |
| A0.8 | 56.7% |

In all the scenarios in which pick-up points are not used, the time available for deliveries is the main limiting factor. Consequently, the results showed that the best vehicle type is always the light van, as the capacity of the larger type cannot be fully used, and the route length duration is more limiting than the physical capacity. In these situations, however, the chosen vehicles were not fully occupied in relation to their load capacity and the reduction in the number of vehicles needed depends mostly on the consolidation of deliveries to a smaller number of stopping points (i.e., more deliveries directed to PPs).

In other words, the use of PPs allows consolidating deliveries to a smaller number of points, which allows each vehicle to make more deliveries at the same time available, thus making it possible to use vehicles with greater load capacity. Figure 10 shows the change in the fleet mix as demand (Q) and the number of deliveries destined for PPs (A) grow.

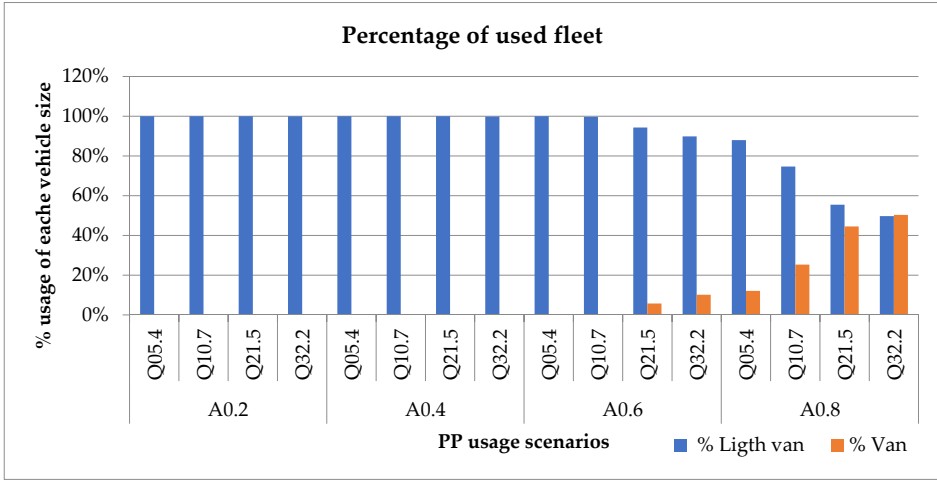

**Figure 10.** Percentage of used fleet.

In scenarios where the percentage of the deliveries destined for PPs are 20% and 40% (A0.2 and A0.4), no changes were observed in the fleet mix; that is, only light vans were used in all the scenarios. In scenario Q21.5_A0.6 (the demand is of 2000 daily packages, and the percentage of the demand delivered through the PPs is 60%), larger vehicles (vans) were assigned to some routes. When 80% of the deliveries are sent to PPs (i.e., A0.8 scenarios), the use of larger vehicles (van) occurs in all the scenarios; in the scenario that corresponds to the higher demand (3000 packages per day, 32.2 packages/km$^2$), the mix fleet is equally distributed between light vans and vans.

Surprisingly, the results also show that motorcycles are not an efficient alternative for such a high-density delivery area as their capacity is very limited, which limits the maximum number of stops on a route. The use of PPs as intermediate transit points (i.e., instead of departing from the distribution center, some last-mile routes may depart from PPs) is not part of the scope of this research, but it may establish an operation in which motorcycles are more suitable.

*5.3. Environmental Impact*

The environmental impact of consolidated deliveries in PPs can be measured by two main factors: the reduction in mileage covered, dealt with in Section 5.1, allows for a reduction in fuel consumption, which is directly related to the emission of polluting gases; and the reduction in the number of vehicles used, representing a potential for reducing traffic in the urban habitat, as well as a reduction in the occupation of parking spaces that might be used by delivery vehicles, as addressed in the above subsection.

The average fuel consumption of the vehicle profiles used in this study is shown in Table 4. For light vans and motorcycles, the fuel considered is gasoline, and for vans, diesel.

**Table 4.** Vehicle fuel consumption.

|  | Motorcycle | Light Van | Van |
| --- | --- | --- | --- |
| Vehicle fuel consumption (km/L) | 35 | 10.7 | 10 |

Fossil fuel consumption is directly related to the emission of pollutants into the atmosphere. In metropolitan regions, emissions of pollutants such as carbon dioxide ($CO_2$), carbon monoxide (CO), nitrogen oxides (NOx), and particulate matter (PM) are widely associated with the use of cars, buses, trucks, and motorcycles. These substances are classified as toxic since they produce negative effects on health when absorbed by the respiratory system.

According to the 2019 vehicle emissions report, prepared by the Environmental Company of the State of São Paulo [35], the pollutant emissions by vehicle profiles used in this study are as shown in Table 5.

**Table 5.** CETESB pollutant emission indicators.

|  | Motorcycle | Light Van | Van |
| --- | --- | --- | --- |
| Year of manufacture | 2020 | 2020 | 2020 |
| $CO_2$ emission (g/km) | 42 | 190 | 235 |
| CO emission (g/km) | 0.834 | 0.215 | 0.032 |
| NOx emission (g/km) | 0.033 | 0.013 | 0.221 |
| PM emission (g/km) | 0.0035 | 0.0011 | 0.0088 |

From the simulation results, it was possible to obtain the total vehicle–km of the delivery operation in all the scenarios studied. Using this information and considering the data shown in Table 4, it was possible to calculate the fuel consumption and consequent pollutant emissions caused by a traditional delivery operation, without using collection points, which are shown in Table 6 according to each demand density.

**Table 6.** Fuel consumption and emission of pollutants for operations without PPs.

|  | Q05.4 | Q10.7 | Q21.5 | Q32.2 |
|---|---|---|---|---|
| Vehicle fuel consumption (thousand L/year) | 17.26 | 30.26 | 54.47 | 76.41 |
| $CO_2$ emission (kg/year) | 47,623 | 80,100 | 140,796 | 195,827 |
| CO emission (kg/year) | 39.76 | 69.70 | 125.46 | 176.00 |
| NOx emission (kg/year) | 2.44 | 4.27 | 7.69 | 10.79 |
| PM emission (kg/year) | 0.20 | 0.36 | 0.64 | 0.90 |

The scenarios where all packages are delivered to their final home destinations (instead of at PPs) were used as a reference to calculate fuel consumption reductions. Reductions in fuel consumption are shown in Table 7.

**Table 7.** Impact on fuel consumption with the use of PPs.

| Fuel Consumption Reduction (Thousand L/Year) | | | | |
|---|---|---|---|---|
| **Scenarios** | **Q05.4** | **Q10.7** | **Q21.5** | **Q32.2** |
| A0.2 | 1.3 | 3.4 | 6.8 | 10.4 |
| A0.4 | 3.7 | 8.2 | 15.6 | 22.2 |
| A0.6 | 6.6 | 13.0 | 23.8 | 33.6 |
| A0.8 | 9.1 | 17.2 | 31.8 | 45.4 |

Again, considering the results for the scenarios in which PPs are not considered (Table 6), we could calculate the reduction in the emission of pollutants in each of the scenarios where collection points are used, shown in Table 8.

**Table 8.** Impact on $CO_2$ emission with the use of PPs.

| $CO_2$ Reduction (kg/Year) | | | | |
|---|---|---|---|---|
| **Scenarios** | **Q05.4** | **Q10.7** | **Q21.5** | **Q32.2** |
| A0.2 | 3241 | 8538 | 16,917 | 25,947 |
| A0.4 | 9328 | 20,557 | 39,089 | 55,657 |
| A0.6 | 16,284 | 32,514 | 59,922 | 84,971 |
| A0.8 | 23,059 | 44,181 | 82,521 | 117,674 |

The results show that it is possible to reduce between 3 and 117 tons of $CO_2$ emissions per year with the use of PPs. The average $CO_2$ reduction for the scenarios A0.2 to A0.8 ranges from 11% to 56%, while for intermediate scenarios A0.4 and A0.6, the reductions are 25% and 40%, respectively.

Similarly, Table 9 shows the reduction (in kg) in CO emission in one year of deliveries for different demands (Q) and the percentage of the demand delivered by the PPs (A) to the delivery model with PPs.

**Table 9.** Impact on CO emission with the use of PPs.

| CO Reduction (kg/Year) | | | | |
|---|---|---|---|---|
| **Scenarios** | **Q05.4** | **Q10.7** | **Q21.5** | **Q32.2** |
| A0.2 | 3 | 8 | 16 | 23 |
| A0.4 | 9 | 19 | 36 | 51 |
| A0.6 | 15 | 30 | 57 | 83 |
| A0.8 | 22 | 44 | 90 | 132 |

In scenarios with greater consolidation of deliveries to PPs (for example, Q32.2_A0.8), it is possible to obtain a reduction of 132 kg of CO emission in one year. This is possible

as fewer polluting vehicles are used: the CO emission of light vans is 0.215 g/km, while for a larger vehicle, such as a van, the emission becomes approximately 7 times lower (0.032 g/km) due to the fuel used by the vans (diesel instead of gasoline). However, for this same reason, the scenarios of greater consolidation also showed an increase in the emission of other pollutants (NOx and PM), as shown in Tables 10 and 11.

**Table 10.** Impact on NOx emission with the use of PPs.

| | NOx Reduction (kg/Year) | | | |
|---|---|---|---|---|
| Scenarios | Q05.4 | Q10.7 | Q21.5 | Q32.2 |
| A0.2 | 0.2 | 0.5 | 1.0 | 1.4 |
| A0.4 | 0.5 | 1.2 | 2.2 | 3.0 |
| A0.6 | 0.9 | 1.8 | 1.0 | −1.0 |
| A0.8 | 0.0 | −2.5 | −13.2 | −22.2 |

**Table 11.** Impact on PM emission with the use of PPs.

| | PM Reduction (kg/Year) | | | |
|---|---|---|---|---|
| Scenarios | Q05.4 | Q10.7 | Q21.5 | Q32.2 |
| A0.2 | 0.02 | 0.04 | 0.08 | 0.12 |
| A0.4 | 0.04 | 0.10 | 0.18 | 0.26 |
| A0.6 | 0.08 | 0.15 | 0.19 | 0.18 |
| A0.8 | 0.06 | 0.02 | −0.28 | −0.52 |

The NOx emission for light vans is 0.013 g/km, but for vans, the emission becomes 17 times higher (0.221 g/km). The PM emission is 0.0011 g/km for light vans and 8 times higher (0.0088 g/km) for vans.

In this way, the implementation of pick-up points makes it possible to reduce the environmental impacts caused by package delivery operations but given a demand (Q) and a percentage of the demand delivered through the PPs (A), it is necessary to use larger vehicles (as vans) that use diesel as fuel, and the environmental impacts can be greater than in a traditional home delivery operation by smaller vehicles with gasoline as fuel.

## 6. Conclusions

With more frequent and less consolidated deliveries, online shopping generates observable impacts, particularly in large, densely populated urban areas, influencing traffic and the environment and constituting one of the most problematic issues for retailers in terms of service and organization costs.

In addition to being an eco-friendlier delivery alternative, PPs help to solve receiving difficulties faced in the traditional delivery model, as they do not require the presence of the receiver, and in some cases, the service is available 24/7, so the customer can collect when it is convenient. Other benefits, such as faster delivery times, lower freight costs, and greater convenience for returning a purchase can be offered by logistics operators to encourage the use of this delivery option.

We proposed an optimization-based modeling approach aimed to investigate the conditions in which a network of pick-up points can be more efficient than home deliveries from both the operational and economic points of view, in the context of urban last-mile distribution. It differs from previous related works found in the literature as we employed optimization techniques instead of analytical models to estimate such impacts as well as to compare PPs with home deliveries for e-commerce shopping. We believe that the potential and benefits of using such an optimization-based approach to address complex real-world urban last-mile delivery problems have been highlighted, as it allows not only a more detailed and precise representation and but also more accurate and realistic outcomes. The location model we propose can be used to determine the minimum number of PPs for

different maximum walking distances, especially in the context of dense and congested urban settings, in which walking is the best or preferred alternative from both individual and public perspectives. Similar benefits derive from the fleet size and mix routing heuristic that allows determining the best fleet mix and distance-efficient routes such that the total delivery costs are minimized.

We have also presented some results of its application in the context of the megacity of São Paulo, Brazil, using operational parameters and data that represent the real operation of the logistics operator in the urban perimeter of the expanded city center region. Several scenarios were evaluated, comprising different densities of deliveries and percentages of deliveries destined for PPs. The results show that PPs can be a promising alternative for reducing the environmental externalities of e-commerce home delivery operations, as fleet and vehicle mileage can be reduced by more than 50% in case most deliveries are destined for these locations.

The results are measured in relation to operational performance, quantifying the vehicle–km, the quantity, and the occupation of the fleet. These operating results are related to the impacts on the environmental efficiency associated with the potential traffic reduction and parking spaces availability, as well as the reduction in fuel consumption, which is directly related to polluting gas emissions.

According to the results we have obtained, it is possible to verify that the implementation of the delivery model using pick-up points, in any scenario, presents a reduction in the mileage traveled and the fleet number compared to the home delivery model and can effectively alleviate externalities related to the traffic in our streets.

The percentage of the demand delivered through the PPs (A) is the parameter that has the greatest impact on mileage and fleet reduction. When the demand delivered through the PPs is 80% (A0.8) of the total demand, the mileage and fleet reductions are always greater than 50% at any demand density, compared with home delivery. Additionally, it mitigates one of the most common problems, that is, failed deliveries (i.e., when the delivery person cannot find a recipient at the destination address). In this case, the package must be returned to the fulfillment center, which yields to additional cost, delay, risk of damage, as well as additional externalities caused by new attempts to deliver. There is also the problem of many apartment buildings that have been unable to receive and handle an increased number of packages as the pandemic has led most residents to spend more at home and choosing to order online instead of venturing out to stores for basic supplies. At many complexes, there may not be enough staff to accept and sign for received packages and then contact tenants about their deliveries. For these cases, lockers can be a solution.

Our results show that the main vehicle used in deliveries is the light van, but in scenarios of greater consolidation, there is a partial change in the fleet profile to vans. The use of vehicles with a smaller profile (e.g., motorcycles), commonly employed in developing countries for deliveries, was not shown to be advantageous due mainly to their low carrying capacity.

The implementation of a PP operation can cause a positive environmental impact, reducing by up to 45,400 L fossil fuel consumption, as well as a reduction of up to 132 kg of CO (carbon monoxide) in a period of one year. However, the results showed that in some cases, in which it was necessary to change the fleet profile to use larger vehicles (that use diesel instead of gasoline), there was an increase of up to 22.2 kg of NOx (nitrogen oxides) and up to 0.52 kg of PM (particulate matter) in the same period.

As seen in the results, PPs represent a promising alternative for reducing the environmental problems of the e-commerce home delivery operation, as well as enabling the reduction of negative impacts that the urban transport of goods cause in urban centers of large cities. A successful implementation of pick-up points depends on demand density factors, distribution center location, fleet characteristics, and other factors addressed in this work.

The framework is flexible and can be used to estimate the potential operational and environmental benefits of implementing the use of PPs not only from the point of view

of a logistics provider but also from an overall socio-economic perspective by public authorities that may be considering measures to stimulate their use as measures to mitigate the curbside, traffic, and environmental externalities of e-commerce's home deliveries.

While analytical models, especially continuum approximation (CA)-based methods that can be used to predict urban route distances, have evolved significantly in the recent years (within 5–15% of near-optimal solutions according to Merchan and Winckenbach [36]), the use of VRP heuristics gives more accurate and detailed results (e.g., number of routes per scenario, number of deliveries per route, route length, vehicle–km, etc.) that an analytical approach cannot provide. Such outcomes can be important to convince some stakeholders, particularly individual shippers and logistics providers that may be interested in offering the convenience of delivering to PPs. This can be particularly relevant in larger and more realistic problem instances, such as the ones that we have addressed, with hundreds to thousands of daily deliveries, in which analytical results cannot be so easily visualized and understood. However, it requires detailed input data that may not be easily available or not applicable in the case of studies that seek global results in whole urban areas. In such contexts, an analytical approach to estimate the outcomes can be more suitable.

Finally, there are relevant factors on the decision to implement e-commerce delivery operations that depend on specific potential future research and that have not been covered in the scope of this work, such as delivery time windows; purchase returns and reverse logistics implemented through PPs; the PP capacities that might allow some deliveries not to be destined for a location as it is momentarily full; and the maximum number of days users have to collect/retrieve their parcels and how this influences both the capacity of the pick-up points as well as the performance of the last-mile delivery system. Taking these aspects into account may probably require an analysis that might need to consider stochastic aspects in modeling. The trade-off between delivery routing costs and the costs of establishing and operating PPs and the use of cars to retrieve packages from PPs by some receivers that may not be willing to walk are other aspects that have not been addressed in the literature. We leave them all as topics for future research.

**Author Contributions:** The authors confirm contribution to the paper as follows: study conception and design: C.B.C.; model conception: C.B.C. and R.M.; model development and implementation: R.M.; data gathering: R.M. and C.B.C.; model simulation: R.M.; analysis and interpretation of results: R.M. and C.B.C.; draft manuscript preparation: R.M. and C.B.C. All authors have read and agreed to the published version of the manuscript.

**Funding:** The first author acknowledges CAPES (Brazil's Coordination for the Improvement of Higher Education Personnel) and the second author Brazil's CCP (National Council for Scientific and Technological Development) for the financial support (grant numbers 88887.513264/2020-00 and 309424/2018-6, respectively).

**Data Availability Statement:** Data generated during the study that support the reported results can be found at the link below. Masteguim, Rhandal (2022), "An Optimization-Based Approach to Evaluate the Operational and Environmental Impacts of Pick-up Points on e-Commerce Urban Last-Mile Distribution: A Case Study in São Paulo, Brazil," Mendeley Data, V1, link: https://data.mendeley.com/datasets/g8hxzzd5v9/1 (accessed on 5 April 2022).

**Acknowledgments:** The authors want to thank Eva Ponce Cueto from MIT Center for Transportation and Logistics for the suggestions and discussions in the early stages of this research. They are also thankful to the anonymous referees for their valuable feedback to improve the quality and clarity of the paper.

**Conflicts of Interest:** The authors declare that they have no known competing financial interests or personal relationships that could have appeared to influence the work reported in this paper.

## Appendix A

**Table A1.** Detailed outcomes of the VRP scenarios analyzed.

| Scenarios | | Number of Routes | | Number of Deliveries per Route | | Km per Route | | CO Emission (kg) | | NOx Emission (kg) | | MP Emission (kg) | | CO$_2$ Emission (kg) | | Fuel Consumption (L) | |
|---|---|---|---|---|---|---|---|---|---|---|---|---|---|---|---|---|---|
| Density | % Demand Destined for PPs | Average | StdDev (%) | Average | StdDev (%) | Average | StdDev (%) | Average | StdDev (%) | Average | StdDev (%) | Average | StdDev (%) | Average | StdDev (%) | Average | StdDev (%) |
| Q05.4 | Without a PP | 9.0 | 0% | 55.6 | 0% | 111 | 1% | 215 | 1% | 13.2 | 1% | 1.1 | 1% | 255,849 | 1% | 93 | 1% |
| | A0.2 | 9.0 | 0% | 55.6 | 0% | 109 | 1% | 211 | 1% | 12.9 | 1% | 1.1 | 1% | 251,223 | 1% | 92 | 1% |
| | A0.4 | 7.6 | 7% | 66.1 | 7% | 110 | 2% | 179 | 5% | 11.0 | 5% | 0.9 | 5% | 216,502 | 4% | 78 | 5% |
| | A0.6 | 6.0 | 0% | 83.3 | 0% | 114 | 2% | 147 | 2% | 9.0 | 2% | 0.8 | 2% | 182,007 | 2% | 64 | 2% |
| | A0.8 | 4.0 | 0% | 125.0 | 0% | 118 | 1% | 101 | 1% | 6.2 | 1% | 0.5 | 1% | 132,859 | 1% | 44 | 1% |
| Q10.7 | Without a PP | 17.0 | 0% | 58.8 | 0% | 106 | 1% | 388 | 1% | 23.8 | 1% | 2.0 | 1% | 443,238 | 1% | 168 | 1% |
| | A0.2 | 15.0 | 0% | 66.7 | 0% | 108 | 1% | 348 | 1% | 21.3 | 1% | 1.8 | 1% | 400,552 | 1% | 151 | 1% |
| | A0.4 | 12.8 | 3% | 78.2 | 4% | 106 | 1% | 293 | 3% | 18.0 | 3% | 1.5 | 3% | 340,302 | 2% | 127 | 3% |
| | A0.6 | 10.0 | 0% | 100.0 | 0% | 108 | 1% | 232 | 1% | 14.2 | 1% | 1.2 | 1% | 273,851 | 1% | 101 | 1% |
| | A0.8 | 7.0 | 0% | 142.9 | 0% | 110 | 2% | 162 | 6% | 14.7 | 69% | 1.0 | 37% | 201,120 | 2% | 72 | 2% |
| Q21.5 | Without a PP | 32.0 | 0% | 62.5 | 0% | 104 | 1% | 715 | 1% | 43.9 | 1% | 3.7 | 1% | 800,719 | 1% | 311 | 1% |
| | A0.2 | 28.2 | 2% | 70.9 | 2% | 105 | 1% | 638 | 2% | 39.1 | 2% | 3.3 | 2% | 716,849 | 2% | 277 | 2% |
| | A0.4 | 22.6 | 2% | 88.5 | 2% | 106 | 1% | 514 | 3% | 31.5 | 3% | 2.6 | 3% | 581,617 | 3% | 223 | 3% |
| | A0.6 | 17.2 | 3% | 116.3 | 3% | 107 | 1% | 395 | 1% | 24.2 | 1% | 2.0 | 1% | 451,996 | 1% | 172 | 1% |
| | A0.8 | 13.0 | 0% | 153.9 | 0% | 106 | 2% | 249 | 5% | 70.7 | 17% | 3.5 | 13% | 339,726 | 2% | 130 | 2% |
| Q32.2 | Without a PP | 46.6 | 1% | 64.4 | 1% | 102 | 1% | 1.026 | 2% | 62.9 | 2% | 5.2 | 2% | 1,139,775 | 2% | 446 | 2% |
| | A0.2 | 40.0 | 0% | 75.0 | 0% | 103 | 0% | 888 | 0% | 54.5 | 0% | 4.5 | 0% | 989,504 | 0% | 386 | 0% |
| | A0.4 | 32.0 | 0% | 93.8 | 0% | 104 | 1% | 714 | 1% | 43.8 | 1% | 3.6 | 1% | 799,200 | 1% | 310 | 1% |
| | A0.6 | 25.0 | 0% | 120.0 | 0% | 104 | 1% | 552 | 3% | 42.9 | 27% | 3.2 | 13% | 630,490 | 1% | 243 | 1% |
| | A0.8 | 18.2 | 2% | 164.9 | 2% | 103 | 2% | 303 | 9% | 140.3 | 22% | 6.3 | 18% | 457,042 | 2% | 179 | 2% |

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
