# Peer review of "An Optimization-Based Approach to Evaluate the Operational and Environmental Impacts of Pick-Up Points on E-Commerce Urban Last-Mile Distribution: A Case Study in São Paulo, Brazil"

_sustainability, doi:10.3390/su14148521_

Round 1

Reviewer 1 Report

This manuscripts proposes a quantitative approach to investigate the suitability of pickup points, comparing them to home deliveries. A model and solving method is proposed and some scenarios assessed to illustrate that framework. The topic is worth of investigation and the paper seems having potential. However, there are some major points for which I think it requires major revisions.

My first concern is related to the literature review and the position and motivation of the research. From the literature review, a non-expert reader would think the paper adresses a subject not really examined in literature. However, there are works (and some of them dating from 10 years ago) on how pickup points and home deliveries substitute shopping trips, so in a way comparing pickup points and home deliveries. For sure they can be outdated or do not consider recent advancements, but the paper proposed here is not entirely novel, so it needs to refer to that literature. More precisely, Durand and Gonzalez-Feliu (2012), Gonzalez-Feliu et al. (2012), Ghajargar et ak. (2016) and Cardenas et al., (2017) address the issue of home deliveries vs pickup points, showing already that pickup points can reduce more than 50% distances than home deliveries. Models used in those works are analytical, not based on OR, so there is a good motivation and position for this paper, but basic works on the subject need to be cited (it is strange to me not seen those works although the aim is similar to that of this paper).

Then, all that leads to the motivation of the paper: knowing that there are already frameworks aiming to estimate the impacts of both home deliveries and pickup points that support decision in a simple, transferrable way, in what the proposed paper adds insight and is novel with respect to the state-of-the-art? In what optimization can add new issues to analytical models? what is the scientific interest of hte proposed framework? And the practice interest? I do not say that the paper does not have a motivation, only that it needs to be better justified. Novelty only is not a necessary condition for a research, but here there is not even novelty (there are other "not presented" works making almost the same). Plesae, present this motivation by referring correctly to the state-of-the-art.

A second major concern is on a methodological issue: the proposed framework is not a simulation-optimization. Reading the paper and more precisely the methodology several times I understand that authors present a combinatorial optimization problem for which a heuristic algorithm is presented, and they assess various scenarios. I do not see the "simulation" in it, which, in computational sciences, will refer to the use of discrete simulation, monte-carlo simulation or system dynamics (or even multi-agent systems) but any of those frameworks are proposed. Simulation means data generation via an authomatic, iterative process (called a simulation process), what I see here is scenario assessment. Those notions are often missunderstood but since "simulation-optimization" is a known concept in operations research (see, for example, Rabe et al., 2021, who apply it to parcel locker location). If scenarios are generated by simulation, explain the way of generating it more properly.

More precisely, Figure 2 needs to be better explained, detailing all assumptions (which are the main variables/parameters of each scenario and why are they assumed (for example, the demand quantities, why and how are they determined; how are they assigned to customers, why and how the pp candidates are defined, etc., see Rabe et al., 2021 for assumtpions of this nature);

The model is a location one, how it is linked then to vehicle routes? In other words, define the assumptions that make the framework pass from the locations of PP to the definition of vehicles and routes (see Gonzalez-Feliu et al., 2012 for those types of assumptions).

Finally the method is a two-stage procedure that first makes a location problem for defining PP locations then solves a VRP for route calculation. This is not clearly explained, please present the method more precisely.

Moreover, it is not clear to me why VRP Spreadsheet solver was not able to solve the problem since, even for more than 200 customers, a zoning can be made and then relate the problem to a set of problems of less than 200 customers. It should be interesting then to compare both approaches. Then another time, authors DO NOT CITE the good source: VRP Spreadsheet Solver is developed by Gunes Erdogan (Erdogan, 2017) and authors do not cite this author in the references. Please, refer correctly to sources to avoid confusion.

The taylor-made heuristic algorithm needs to be described. If it is developed in another work, cite it, if not define it. If it is used for a location problem.

Table 1 values (costs) need to be justified (from where authors do obtain them?)

Why authors estimate CO when the main environmental indicator is CO2? I think authors need to first make CO2 estimations for global warming, then, if they want to add pollution indicators, PM10 and then CO, NOx and SOx. But only CO is not usual and not relevant (either CO2 or a set of environmental indicators, includinc CO2, CO and PM10 at least).

References
  • Cardenas, I. D., Dewulf, W., Vanelslander, T., Smet, C., & Beckers, J. (2017). The E-Commerce Parcel Delivery Market and the Implications of Home B2C Deliveries Vs Pick-Up Points. International Journal of Transport Economics, 44(2).
  •   Durand, B., & Gonzalez-Feliu, J. (2012). Urban logistics and e-grocery: have proximity delivery services a positive impact on shopping trips?. Procedia-Social and Behavioral Sciences, 39, 510-520.
  • Erdoğan, G. (2017). An open source spreadsheet solver for vehicle routing problems. Computers & operations research, 84, 62-72.
  • Ghajargar, M., Zenezini, G., & Montanaro, T. (2016). Home delivery services: innovations and emerging needs. IFAC-PapersOnLine, 49(12), 1371-1376.
  • Gonzalez-Feliu, J., Ambrosini, C., & Routhier, J. L. (2012). New trends on urban goods movement: modelling and simulation of e-commerce distribution. European Transport, 50(6), 1-23.
  • Rabe, M., Gonzalez-Feliu, J., Chicaiza-Vaca, J., & Tordecilla, R. D. (2021). Simulation-Optimization Approach for Multi-Period Facility Location Problems with Forecasted and Random Demands in a Last-Mile Logistics Application. Algorithms, 14(2), 41.

Reviewer 2 Report

This paper describes a modeling approach to investigate the conditions in which a network of pickup points can be more efficient than home deliveries from operational and environmental points of view. 

The paper addresses a relevant research topic on urban last mile logistics and tackles a severe challenge of urban transport and mobility systems.

However, several aspects of the paper require amendments before publication:

  • Since the results of the vehicle routing problem are an integral part of the methodology, the tailor-made heuristic requires further description and explanation.
  • The assumptions on fleet costs and operational parameters assumed in Table 1 require substantiation by explanations and references.
  • Units of measurements should be made clear; e.g. Table 2: vehicle-mileage (km) per day.
  • The results shown in Table 2 require further explanations. The sentence in line 436/ 437 is not sufficiently comprehensible.
  • It is furthermore not clear, how the fleet size has been computed.
  • In terms of environmental impacts, it is highly recommended to assess the impacts on CO2 emissions, too.
  • The conclusions require a critical review of the method, results and underlying assumptions (e.g., is it a realistic assumption that residents do not use own car or motor-bike for the ride to the pickup points?).
  • Finally, the variable MinPerc is not yet used consistently (MinPerc vs. PercMin).

Reviewer 3 Report

The paper is well written and addressed important issues regarding the impact of e-commerce deliveries on environments. Simulation results show that pick up points provide better environmentally friendly alternatives.

Even though it's clear on the overall picture, could the author comment on how to persuade stakeholders in the system to be more interested in using PP? For example, the recipients who might feel more convenient waiting at home or the motorcycle riders whose income might be reduced.  

Round 2

Reviewer 1 Report

This paper is a reviewed version of an already submitted paper. I see authors made an effort to improve it and the result is qualitative. Howeer, I see still some minor points needing to be adressed before final acceptation:

1. Authors do not refer to basic (mainly recent) works on VRP for e-commerce optimization. Moreover, I think the foundations of e-commerce and city logistics interactions need to be referred properly (i.e. Nemoto, 2003; Gonzalez-Feliu et al., 2012; Hayashi et al., 2014; Visser et al., 2014). They set some issues such as substution of shopping trips by e-commerce trips (also adressed in other works cited by authors) as well as the environmental issues of those switches.

2. Moreover, and although they are not proposing an algorithm for pure optimization purposes but for assessement uses, it is important to refer to proper litterature. I understand that authors cannot report all works on the field, but since they are addressing VRP in e-commerce last mile with urban logistics e-commerce assessment they need to address both types of works. And there are some recent works with VRP algorithms for e-commerce, authors need to position their algorithm with respect to them and show in what the proposed work is different in form and uses (I am convinced their work is more suitable for assessment than those of published papers, but authors cannot "hide" those papers but refer the proposed work to the state-of-the-art on VRP for e-commerce. Examples of such works (after a rapid search and a relevance control to be sure that they are suitable and related to the authors' paper) can be (but maybe they are others): Moons et al. (2018), Prajapati et al. (2020), Özarık et al. (2021), Van Woensel and Laporte (2021), Vincent et al. (2022) and for a review on VRP in e-commerce last mile deliveries Archetti and Bertazzi (2021).

3. The fact that a type of methos has not been used for an assessment can be interesting but alone it does not make a sufficient condition to make a research relevant. I am not sure this is the first work using VRP to compare home deliveries and pickup points (maybe), but even on that, as I already said, there are works (mainly Durand and Gonzalez-Feliu, 2012 and an in-depth methodological description of the models and assessment methods in Gonzalez-Feliu, 2018) using analytical models. So it is not the first time an assessment of this type of scenarios is made (other authors have therefore used also analytical models to those purposes) and those works are applicable and able to generate distances and therefore CO2 rates and other pollutants, plus eventually economic costs and benefits (Gonzalez-Feliu, 2018). So, why a VRP will add insight and new issues to already existing models? I am not saying the work is not adding insight, I ask only authors to say clearly in what a VRP approach can complete or be more suitable than analytical models for assessment issues. For example, analytical models are quick and give estimates of highly aggregate results, but finally all routes have more or less the same composition. VRP can add heterogeneity and variability to routes and eventually (if using VRP-TW, show the influence of time in route construction (Ando and Taniguchi, 2005, Deflorio et al., 2012). This is only a first idea but I think authors know exactly what this VRP estimation adds to current analytical models (as Nemoto et al., 2001; Durand and Gonzalez-Feliu, 2012; Gonzalez-Feliu et al., 2012; Miyatake et al., 2016).

4. When presenting the methodology, and mainly the algorithm, authors need to better justify the algorithm used. There are some works on VRP that show the insterest of some heuristics instead of others. For example, Laporte and Semet (2002) show the interest of using classical heuristics instead of meta-heuristics. The most complete and recent review of VRP for city logistics (which includes considerations on B2C flows also in e-commerce last mile transport), i.e. Cattaruzza et al. (2017), and that of e-commerce VRP, i.e. Archetti and Bertazzi (2021), are basic works that need to be cited properly. I think that although authors cited some works, they are still "forgetting" previous work and the paper as currently written gives the impression the method authors use is new, novel or innovative when it is a standard one. It is the type of results and the use of a standard algorithm, simpler than advanced metaheuristics, but which does not blocks on real-size instances, as an assessment tool, which is interesting (but not new) and the scenarios/assessments/results itselfs which constitutes the paper novelty. So please, refer to basic works on VRP (those three but also others if needed) to position the method as a quick heuristic (for sure, neither one of the simplest ones but nor a complex "black box" metaheuristic algorithm).

5. I am not convinced on explanation in p. 11 about demand. A major e-commerce retailer's demand (which general volumes? for which city/cities?/For how many months/years you considered the demand? Is it only a delivery day/month or a period of time in which average/median/bayesian generation rates can be defined?) and cutting off some demand (because it cannot go to the lockers) is not enough. Concerning the assumption of cutting of big parcels: which percentage do that represent? Why can we assume they are not in your simulation? Is it because the carrier manages them separately? (i.e. 2 types of routes: big demands and small demands) If yes, provide evidence, if not, it is not possible to cut off this, you need to do otherwise: including big demands in home delivery or shop routes compulsory and when transferring demand to lockers taking only small demands. Cutting off what you do not like or what makes you difficulty is an easy way of biasing estimations, since it does not represent a reality (Ackoff, 1977) and makes me think the scenarios are not coherent with a real application, so results would loose their interest.

6. Now the case presentation is clearer but I still miss some precisions on the way data has been collected from the field: how many interviews have been made? Which are the main companies (sector, size, not necessarily names)? What data has been asked? (a demand on an X months period, the main fixed and variable costs, etc.), same for fuel consumptiona nd environmental emissions: which are the sources? Interviews, a model (like COPERT in Europe) or other documents? Please, precise for each parameter/input data the origin (either the field data collection, in that case it is important to put, it can be a table, the data asked in the interviews and if it an average value, a file or other data types and the period, if it is one day, one month, and so on, then how the presented individual paramleters are assumed/calculated). Authors added some explanations, which is really appreciable, but I think not all of them are correcly adressed. In other words, and if I did not miss something, the source of Table 4 is missing (ok, they are average consumptions, but from which data are they obtained?), and this is often seen in the paper. Each parameter/input needs to be justified, either by a document/reference or a calculation method (a dataset obtained by interviews/from companies, with a small explanation of from whom and how these data are obtained, or in a worst case the result of a consensual focus group or practitioners forum, but a value without a justification cannot feed a research paper, in my advice). If authors did not have a reliable source for some parameters or inputs, they can make an assumption, but in this case they need to justify it, maybe by a focus group of an informal exchange with a practitionner and a statement that the value remains the same for all scenarios so if there is a bias it is the same for each scenario.

7. Results show that PP allow to reduce travelled distances (result already shown with analytical models in Gonzalez-Feliu et al., 2012 and Durand and Gonzalez-Feliu, 2012, among others). That distance reduction leads to a fuel consumption reduction and to environmental improvements. But those issues can be obtained with analytical models. What makes the specificity of using VRP is, in my experience, two main issues: the first is that VRP are company-based, so they can be used at a company level (analytical models can also be applied to companies, but since they are aggregated they are more suitable for global results in whole urban areas for which input data are lower); VRP can give more accurate results than analytical models at company level if input data are known but need the physical (and eventually temporal) distribution of demand: if this is known, I think VRP (at company level were demands are from hundreds to thousands) are suitable approaches and if not, or at a global level (with tens of thousand demands and multiple companies), analytical models are more suitable. That discussion needs to be extended either in results or conclusion section.

8. The second issue is that VRP allows to give more detail in data characterizing routes: you can extract from VRP outputs the number of routes per scenario the number of deliveries per route (average and standard error), if expected the total time per route (average and standard error) and the same for emissions and fuel consumption. This type of results are really interesting and authors can provide them since they are issued from VRP outputs. Authors make already analysis on number of routes and average distances, fuel consumption and emissions, but should provide also the standard errors to show if increasing demand leads to more homogeneous routes. This should be a great result that an analytical model cannot give.

Finally, I think authors need still to improve the paper but are close to the final point. Therefore, I qualify the required revisions as minor ones.

Suggested references:

Ackoff, R. L. (1977). Optimization+ objectivity= optout. European Journal of Operational Research, 1(1), 1-7.

Ando, N., & Taniguchi, E. (2006). Travel time reliability in vehicle routing and scheduling with time windows. Networks and spatial economics, 6(3), 293-311.

Archetti, C., & Bertazzi, L. (2021). Recent challenges in Routing and Inventory Routing: E‐commerce and last‐mile delivery. Networks, 77(2), 255-268.

Cattaruzza, D., Absi, N., Feillet, D., & González-Feliu, J. (2017). Vehicle routing problems for city logistics. EURO Journal on Transportation and Logistics, 6(1), 51-79.

Deflorio, F. P., Gonzalez-Feliu, J., Perboli, G., & Tadei, R. (2012). The influence of time windows on the costs of urban freight distribution services in city logistics applications. European Journal of Transport and Infrastructure Research, 12(3), 256-274.

Gonzalez-Feliu, J. (2018). Sustainable Urban Logistics: Planning and Evaluation. John Wiley & Sons.

Gonzalez-Feliu, J., Ambrosini, C., & Routhier, J. L. (2012). New trends on urban goods movement: modelling and simulation of e-commerce distribution. European Transport, 50(6), 1-23.

Hayashi, K., Nemoto, T., & Visser, J. J. (2014). E-commerce and city logistics solution. City logistics: Mapping the future, 55.

Laporte, G., & Semet, F. (2002). Classical heuristics for the capacitated VRP. In The vehicle routing problem (pp. 109-128). Society for Industrial and Applied Mathematics.

Miyatake, K., Nemoto, T., Nakaharai, S., & Hayashi, K. (2016). Reduction in consumers’ purchasing cost by online shopping. Transportation Research Procedia, 12, 656-666.

Moons, S., Ramaekers, K., Caris, A., & Arda, Y. (2018). Integration of order picking and vehicle routing in a B2C e-commerce context. Flexible Services and Manufacturing Journal, 30(4), 813-843.   Özarık, S. S., Veelenturf, L. P.,

Nemoto, T. (2004). An experimental cooperative parcel pick-up system using the Internet in the central business district in Tokyo. In Logistics Systems for Sustainable Cities. Emerald Group Publishing Limited.   Nemoto, T., Visser, J., & Yoshimoto, R. (2001). Impacts of information and communication technology on urban logistics system. Graduate School of Commerce and Management, Hitotsubashi University.

Prajapati, D., Harish, A. R., Daultani, Y., Singh, H., & Pratap, S. (2020). A clustering based routing heuristic for last-mile logistics in fresh food E-commerce. Global Business Review, 0972150919889797.

Van Woensel, T., & Laporte, G. (2021). Optimizing e-commerce last-mile vehicle routing and scheduling under uncertain customer presence. Transportation Research Part E: Logistics and Transportation Review, 148, 102263.

Vincent, F. Y., Susanto, H., Jodiawan, P., Ho, T. W., Lin, S. W., & Huang, Y. T. (2022). A simulated annealing algorithm for the vehicle routing problem with parcel lockers. IEEE Access, 10, 20764-20782.

Reviewer 2 Report

The paper has been improved significantly and the comments have been considered. Following remarks:

- As concerns the costs (Table 1), it is still not clear how the vehicle purchase costs have been derived (is there any reference, which you have used?), and where the percentage shares of cost components come from.

- The mileage is referred to as "distance travelled ) (5.1) and "km" (Table 2). Wouldn't it be more appropriate to name it "vehicle-km"?

- A further English language check is recommended. E. g.,

- Differently from the related literature, we in which analytical models
were employed, we use optimization and algorithms to determine the economic and environmental benefits of packages destined to pick-up points instead of home deliveries.
